# Development of Ion Character Property Relationship (IC-PR) for Removal of 13-Metal Ions by Employing a Novel Green Adsorbent *Aerva javanica*

**DOI:** 10.3390/molecules27238213

**Published:** 2022-11-25

**Authors:** Fozia Batool, Ali Irfan, Sami A. Al-Hussain, Eida S. Al-Farraj, Shahid Iqbal, Jamshed Akbar, Sobia Noreen, Taslim Akhtar, Tunzeel Iqbal, Magdi E. A. Zaki

**Affiliations:** 1Department of Chemistry, University of Sargodha, Sargodha 40100, Pakistan; 2Department of Chemistry, Government College University Faisalabad, Faisalabad 38000, Pakistan; 3Department of Chemistry, College of Science, Imam Mohammad Ibn Saud Islamic University (IMSIU), Riyadh 13623, Saudi Arabia; 4Department of Chemistry, University of Education Lahore, Jauharabad Campus, Lahore 41200, Pakistan; 5Govt. Associate College for Women, Mandi Bahauddine 50400, Pakistan; 6The Rawalpindi Women University Rawalpindi, Rawalpinfi 46000, Pakistan

**Keywords:** adsorption, aqueous media, green adsorbent, *Aerva javanica*, covalent index, ion charge, IC-PR, polarizability

## Abstract

The novel *Aerva javanica* absorbent was applied for the removal of thirteen selected metal ions from a distilled water solution of each metal by the batch adsorption method. The optimization remediation parameters of the metal ions for the batch adsorption approach were developed, which were the initial concentrations (60 ppm), contact time (60 min) and pH (7). The basic properties of metal ion affected the adsorption results; therefore, 21 properties of metal ions were selected, which are called “descriptors”. The most significant descriptors were selected that were vital for the adsorption results, such as covalent index, polarizability and ion charge. The developed model equation by the descriptors provided more than 80% accuracy in the predicted results. Furthermore, Freundlich and Langmuir adsorption models were also applied on the results. Constants of the Freundlich and Langmuir models were also used for model generation, and the results revealed the importance of a covalent index for the removal phenomenon of metal ions. The current study provided a suitable Ion Character Property Relationship (IC-PR) for the removal of metal ions, and future predictions can be achieved on the proposed adsorbent with significant accuracy. The ecofriendly and cost effective *Aerva javanica* absorbent in the batch experimental model of the current study predicted that this novel absorbent can be used for the removal of a wide spectrum of heavy metal ions from different sources of waste waters.

## 1. Introduction

The natural occurring environmental resources are produced, recycled and controlled by nature itself. Man exploited natural sources to fulfill his necessities, but due to the ever-increasing population, man has been compelled to prepare naturally occurring materials by artificial ways [1,2]. Unfortunately, natural resources are being consumed and deleted by the growing population, with forests burnt and cut to clear them away to make factories, mills, industries and housing places [3,4]. When the natural environment is being disturbed, then the amount of natural resources is also intruded on accordingly. In order to bring sustainability to life, the environmental resources must be reserved [5,6,7]. Heavy and trace metals removal is very significant due to their potency of toxicity for the environment and water bodies’ living creatures and inland living organism. The most severe and most threating issue for today’s world is water pollution, which has attracted the attention of the modern world due to shortages of clean water resources [1,2]. The increasing number of industries and traffic exhausts are the major sources of anthropogenic inputs in fresh water supplies [3,4]. Heavy metals such as Cu, as well as Cd, are cumulative poisonous materials that are accounted to be exceptionally toxic and causes of serious environmental degradation [5,6,7]. All countries in the world seek natural materials for the removal of toxic heavy metals from aqueous media. Natural substances for the removal of heavy metals are largely accessible, commonly by agricultural wastes. These are low-cost materials that are used for the separation of toxic heavy metals and are also environmentally friendly [8]. In industries, chemical techniques are used for the removal of heavy metals [9,10,11]. One of the most destructive drawbacks of these methods is that they are not economically favorable. The adsorption process is the best method for the removal of toxic heavy metals, and it is also an economically cost-effective way [12,13,14]. Different agricultural wastes such as sawdust, tree barks, rice husk, pine bark, animal bones, activated charcoal, etc. are economically low-cost materials that are being used as adsorbents for the separation of the toxic heavy metals. Activated charcoal is largely used as an adsorbent for the removal of toxic heavy metals [15,16,17,18]. Nowadays, too much commercial use of activated charcoal increases the price, so the activated charcoal as a pollutant removal is not economically executable [19,20]. In the current study, we explore a new sorbent for the efficient removal of metal ions from aqueous media. *Aerva javanica* is commonly called desert cotton, which is widely distributed and available in Pakistan. It is a soft and fibrous material in its structure, which is the best suitable option for its utilization in the adsorption process. The action and mechanism of biosorption is highly complex and hard to make a model from and imitate using conventional mathematical modeling. The resulting relationship of the sorption process is highly nonlinear due to the interaction of more variables [21,22]. The Ion Character Property Relationship (IC-PR) is an advanced way to relate the adsorption phenomenon of agro-waste with the structure of the sorbate. In IC-PR, different properties of the ions are related with the adsorption process to generate a theoretical model for the future prediction of the adsorption of metal atoms on a proposed adsorbent [23]. Adsorption is a physicochemical process, and it is mainly based on different chemical properties such as polarity, charge of the ion and its covalent index. It is reported that the covalent index strongly affects the metal binding potential over the surface of the adsorbent, since, as the covalent index increases, the attachment power of metal ions over the adsorbent also increases [24]. Similarly, ions with different charges have variable adsorption potential. It is a basic prerequisite to relate these properties of metal ions with the adsorption property to develop a true picture of the metal attachment on the adsorbent surface and study the underlying phenomenon. Therefore, IC-PR is a suitable way to generate a relationship between adsorption phenomenon and properties of the elements [25]. In the current study, we performed the adsorption of 13 selected metal ions on novel sorbent *Aerva javanica* and developed IC-PR between metal ions and the proposed adsorbent. The adsorption study for all these metal ions was performed in the laboratory to obtain the optimized conditions for the adsorption parameters. Based on these experimental results, a model was generated to correlate the adsorption of metal ions with different properties [26]. The developed model can be successfully applied on all metal ions with a high accuracy and robustness.

## 2. Results and Discussion

### 2.1. Characterization of Sorbents

Before adsorption, the experiment surface of the adsorbent was analyzed for different chemical moieties present on the surface of the adsorbent, which are responsible for the attachment of metal ions on it. The infrared analysis was performed for said purpose.

#### 2.1.1. Fourier-Transform Infrared Spectroscopy

The surface of the adsorbent was examined using a Fourier-Transform Infrared Spectrometer (Model Shimadzu AIM-8800). Different sorbates adhere to the surface of the sorbent due to the presence of certain functional groups. The study of these functional groups reveals information on organic functional groups that are involved in adsorption [27,28,29]. In the range of 4000–450 cm^−1^ wavenumbers, FTIR spectra of the sorbent material were collected; the results are present in Figure 1. It appears that a broad band develops in the 3000–3700 cm^−1^ range, which may be attributable to the –OH stretching vibration of hydroxyl functional groups, including hydrogen bonding. Hydrogen bonding is indicated by the presence of a broad-banded –OH group in this frequency range. The hydroxyl group is one of most important functional groups present on the surface of the adsorbent, as it is responsible for the attachment of sorbate on the surface. The sorbent has a stretching band of –CH in the 2900–3000 cm^−1^ wavenumber range. The C=O peak appears at 1750 cm^−1^, while the C-O functional group peak appears at 1200 cm^−1^. The –CN stretching peak can also be seen in spectra at a value of 1049 cm^−1^. At 1645 cm^−1^, vibration caused by secondary amides can be seen. A peak at 1246 cm^−1^ appears due to C-O of the carbonyl functional group. the carbonyl functional group also plays a role in metal ion attachment on the surface [30]. The –OH group and heteroatoms play an important role in attaching the sorbate to the surface for adsorption [31]. Attachment of metal ions on the surface of the adsorbent occurs mainly due to electrostatic forces developed between metal ions and carbonate functional groups present on the surface of the adsorbent. As far as the complexation attachment of ions to the surface of the adsorbent is concerned, atoms such as nitrogen and oxygen are involved, as they share an electron pair for said purpose. Therefore, –OH groups as they appear in FTIR spectra are responsible for this attachment [32]. The information obtained through a functional groups analysis helps us to understand possible mechanisms of adsorption. The FTIR spectra of all metal ions were also obtained after adsorption, and changes in position of the functional groups confirm the attachment of metal ions on the adsorbent surface [33].

#### 2.1.2. Batch Adsorption for Parameters Optimization

In this experimental work, different parameters were optimized to get the best adsorption conditions for the remediation of 13 selected metal ions on the adsorbent. For this purpose, the contact time was varied from 30 to 120 min, and the best adsorption was achieved at 60 min, as shown in Figure 2. Initially, by increasing the contact time, a significant rise in adsorption was found for almost all metal ions, but further increases in the contact time have less impact on the parentage adsorption of metals. This factor indicates that the active sites available on the *Aerva javanica* surface are occupied by the metal ions, and no more vacant places are available for metal attachment [34]. 

Another important parameter is the pH of the solution while performing adsorption. The pH range was set from 2 to 12, with 1 g of the agro-waste material and 60 min of contact time. The results given in Figure 3 indicated that the adsorption capacity is not significant in the acidic pH range. The adsorption capacity has a positive impact by increasing the pH of the solution. This is because of the fact that, at a low pH, H^+^ ions are present in the solution that compete with metals ions for active sites occupation, so the adsorption is low; when the pH was increased, an increase in the adsorption was observed from the 6 to 8 pH range. Further increases produced metal hydro oxide, and again, low adsorption capacity was achieved [35].

The initial concentration of sorbate has a dominant affect during the adsorption phenomenon. When the concentration was increased, an increase in the adsorption capacity was observed for almost all metal ions. However, further increase in the concentration shows no significant change in the adsorption, as the active sites are already occupied by metal ions [36]. Although, for some metal ions, an increase in the adsorption capacity was observed, but it was not as significant, and the results are given in Figure 4. Previous studies also reported that the accommodation for sorbates decreases as the concentration is very high due to the unavailability of resident sites [37].

#### 2.1.3. IC-PR of Metal Ions with *Aerva javanica* as Adsorbent

*Aerva javanica* (Desert cotton) was used as an adsorbent to investigate the relationship between distinct metal ion characteristics and their adsorption behavior on the sorbent’s surface. Table 1 summarizes the findings of a regression analysis done by Solver in Microsoft Excel to construct a model equation showing the relationship between each parameter and the percent sorption. The relationship between electronegativity, density, atomic radius/atomic weight, standard reduction potential and covalent index, respectively, was found to be remarkable for X5, X14, X15, X18 and X21. Metal adsorption on the adsorbent requires the use of a covalent index and electronegativity, two key properties. Binding forces are responsible for attaching metals to sorbent surfaces, and these descriptors have a direct impact on them. In the same way, the adsorption potential and density of metals help with their binding. These five descriptors were used to create a model equation that is depicted as Equation (1):%Sorption = 72.37606 + 1.318314 × 5 + 0.818116 × 14 + 11.5886 × 15 − 1.34948 × 18 + 2.484041 × 21(1)

This model equation, on the other hand, only achieved 67% adsorption. It was found that using Minitab 17 to run a stepwise multiple regression analysis gave the following results as mentioned in Equation (2):%Sorption = 76.56 − 10.95 × 17 + 3.54 × 20 + 4.002 × 21(2)

(N = 13, R-square = 91.79, predictive R^2^ = 0.8366).

A % sorption of 84.34 was achieved with only these three descriptors, with a R-sq value 91.79 and level of significance α = 0.05. The significant descriptors in this equation were the covalent index (X21), polarizability (X20) and ratio of ion charge and square of the ionic radius (X17). 

These three descriptors have found a prevailing effect on the adsorption phenomenon. The covalent index is an important descriptor that affects the adsorption rate in terms of the metal ion attachment on the surface of the adsorbent. The covalent index of metal ions is a measure of its capacity to exhibit a covalent interaction with the ligands [38]. The higher the covalent index of a metal ion, the more its adsorption on the surface groups of adsorbents. The importance of this descriptor for IC-PR analysis is described by many other researchers as well [24,39,40]. Similarly, cationic polarizability is one of the most frequent forces expressed in binding events. Since nucleophilic and electrophilic sites available on the adsorbent surface and hydrogen binding are directly affected by the cationic polarizability of ions, metals binding on these active sites decides the significance of adsorption for the particular purpose. Metals with elevated levels of cationic polarizability were found prevailing in adsorption phenomenon as compared with the metals of low polarizability [41,42]. 

The size of the ions is another factor playing a role in the adsorption of metal ions. The smaller the packing of ions in the form of hard spheres, the better their attraction for the adsorbent site [43,44,45]. Furthermore, a small size facilitates the adsorption of more ions on the surface of the adsorbent. The ratio of ionic charge and square of the ionic radius were also found as influencing parameters for the present study due to the small sizes of the ions. A probable mechanism of binding is explained by the information present in the IC-PR equation. The underlying mechanism of the binding process can be revealed by careful study of the descriptors. 

The parameters for the Langmuir and Freundlich isotherms were also determined by varying the metal ion concentrations, and the results are given in Table 2. The Freundlich adsorption isotherm was found more affective for the present study, with a high value of R-square (r^2^ > 0.9) as compared to Langmuir. The Freundlich adsorption capacity K_F_ was found above 2, which indicates the significant adsorption efficiency of *Aerva javanica* for all metal ions [46,47,48]. Therefore, *Aerva javanica* gives multilayer adsorption of metal ions with a maximum adsorption capacity for lead (10.05) and minimum for potassium ions (2.01). 

#### 2.1.4. Adsorption Capacity of *Aerva javanica* through Langmuir and Freundlich Isotherm

The adsorption capacity calculated through Langmuir constant Q was used for a regression analysis of metal ions, and the results are present in Table 3 Only a descriptor that gives R-square values above 50% was the covalent index (X21), and the model Equation (3) generated was:Q = −57.8 + 36.4 × 21(3)

Poor results for the relationship of the Langmuir constant represent that, in the present study, monolayer adsorption was suppressed by the multilayer adsorption phenomenon [49,50,51]. A regression analysis through the adsorption capacity generated by the Freundlich isotherm is given in Table 4. Significant descriptors were X21, X14 and X5, respectively, which are in good agreement with the descriptors calculated by % adsorption, as shown in Equation (4): KF = −0.57 + 1.841 × 21(4)

The model equation for the Freundlich adsorption capacity was generated at a significance level 0.05, and the R-square was 79.20%, as depicted in Table 4. The covalent index alone can perform significant adsorption of the metal ions on *Aerva javanica,* as represented later shown in Table 5. The small *p*-value and standard error obtained for a regression analysis of the Freundlich adsorption capacity with metal ions reflects the applicability of this adsorption isotherm on the present study.

#### 2.1.5. Model Validation

In IC-PR development, the real challenge is to develop such a model that can successfully predict the activity of new compounds. Application of the model in this regard is judged by its validation both internal as well as external. A satisfactory internal validation gives permit to the model for applicability on training and test sets, and external validation shows its approach for the prediction of new compounds activity [22,52,53]. For the present model, internal validation was observed by the Q^2^ value, and for external validation, the predictive R^2^ was calculated. A high value of Q^2^ (0.8769) reflects good internal validation, and R^2^ was 0.8366, showing predictive ability of the model for new molecules.

## 3. Materials and Methods

### 3.1. Materials

*Aerva javanica* (Desert cotton) was gathered from various locations in the Sargodha District of Pakistan. Following collection, the sample was thoroughly cleaned with deionized water to eliminate any dust or surface contaminants that had accumulated. In the beginning, the sorbent was dried in an open container at ambient temperatures and then in an electric oven (Model, LEB-1-20) at 105 degrees Celsius for 24 h to completely remove all of the moisture content. The dried sorbent was separated and processed separately, with the proper particle size (10-mesh) being separated by sieves and stored for subsequent examination.

### 3.2. Chemicals

All of the reagents, solvents and chemicals utilized in this study were of analytical reagent grade, and they were purchased from Sigma-Aldrich (Darmstadt, Germany) or Merck (Darmstadt, Germany). To make working solutions, a series of dilutions with double-distilled water was performed on the standard solutions. Analytical reagent grade metal salts were used for the adsorption experiments as well.

### 3.3. Characterization of Sorbent

Characterizing the sorbent is required to determine the various physical and chemical characteristics that influence the adsorption. 

### 3.4. Fourier-Transform Infrared Spectroscopy (FTIR)

A Fourier-Transform Infrared Spectrophotometer was used to identify functional groups in the sorbent’s structure (Model Shimadzu AIM-8800). The detection of these functional groups aids in the identification of the nature of the binding interactions between the sorbate and sorbent surface that cause the sorbate to be adsorbed on its surface [28]. It was decided to employ the diffuse reflectance infrared technique (DRIFT) and use KBr as the background reagent.

### 3.5. Parameter Optimization by Batch Adsorption Experiment

Batch adsorption experiments with sorbate aqueous solutions were used to identify the optimum removal conditions. For this, a standard solution of 100 mL of each metal was used in a 250-mL Erlenmeyer flask with a known sorbent amount, shaking speed and agitation time. The sorbate concentration (30–90 mg/L), pH (3–12) and contact time (40–120 min) were studied as optimizable parameters. After shaking for a particular time, sorbent samples were taken at appropriate intervals (40–120 min) and filtered to remove contaminants. The parameters were set one by one, and the sorption phenomenon was observed in order to determine the parameters that were most appropriate for adsorption [54]. An optimized set of parameters obtained from the first step of the analysis were used for the Ion Character Property Relationship (IC-PR).

### 3.6. Adsorption Study

To evaluate the removal efficiency of the sorbent, filtered sorbate solutions (as mentioned above) were subjected for analysis. To analyze the residual amount of metal ions in a solution, the relevant solutions were subjected to FAAS (AA 6300, Schimadzu, Japan). The fuel and oxidant used were acetylene and air, respectively, and background correction was performed by a Deuterium Discharge Lamp. The sorption of metal ions was calculated as shown in Equations (5) and (6), respectively [55].
(5)$$\%\mathrm{Sorption}=\frac{\mathrm{Ci}-\mathrm{Ce}}{\mathrm{Ci}}\text{×}100$$

Equation (5) depicts the linear Langmuir model, where Ce and Cads are the equilibrium and adsorbed concentrations, respectively. The adsorption capacity is represented by Q, a constant, while the adsorption energy is represented by **b**.
(6)$${\mathrm{LogCad}=\mathrm{LogK}}_{F}+\frac{1}{n}{\mathrm{logC}}_{e}$$

K_F_ shows the adsorption capacity, while 1/n represents the adsorption intensity in Equation (6), which is the linearized form of the Freundlich isotherm.

### 3.7. Selection of Descriptors

For IC-PR, it is critical to use descriptors that accurately capture the unique properties of the elements. There were 21 descriptors chosen to connect the adsorption properties with the metal ionic characteristics. As you can see in Table 1, these descriptors and their respective symbols are listed. We first studied the implications of these descriptors for the attachment of metal ions on the surface of the adsorbent and then selected important descriptors to develop the QSPR model. The interaction of elements with other compounds is mostly determined by their physical and chemical properties, which are chosen as descriptors. From these selected descriptors, we select those that only show the maximum interaction for the purpose [56].

### 3.8. Ion Character Property Relationship (IC-PR)

The ion character property relationship was developed between the adsorption phenomenon and metal properties based on these descriptors. Several software packages, including Minitab 17 and Microsoft Excel 2010, were used in the development of the relationships, as well as the performance of the regression analysis. Using Minitab 17, the correlation coefficient, F-statistics, t-statistics, *p*-value and R^2^ adjusted were calculated and predicted by putting the level of significance at 0.15. The stepwise selection of the descriptors was carried out in order to identify the best fitting model, and then, the single-step analysis was carried out using the descriptors that were selected. These descriptors were utilized in the development of the regression equation.

### 3.9. Regression Equation

Stepwise regression was also performed using Microsoft Excel 2010 tools to calculate the standard error and model equation for each descriptor, as depicted in Equation (7):%Sorption = 15.90 + 24.61 × 6 +82.9 × 16 + 4.087 × 19 + 0.903 × 21 + 9.523 × 22(7)

In Table 5, the characteristics of 13 metals ions are given to develop correlation:

**Table 5 molecules-27-08213-t005:** Metal ion characteristics used to develop a correlation [57,58,59,60].

Symbol	Properties	Ca^2+^	Cr^3+^	Co^2+^	Cu^2+^	Cd^2+^	K^1+^	Mg^2+^	Mn^2+^	Na^1+^	Ni^2+^	Pb^2+^	Zn^2+^	Fe^2+^
X1	Atomic Number	20	24	27	29	48	19	12	25	11	28	82	30	26
X2	Atomic Weight	40.07	51.99	58.93	63.54	112.41	39.09	24.3	54.93	22.98	58.69	207.2	65.39	55.84
X3	Atomic Radius (Å)	1.92	1.25	1.25	1.35	1.48	2.27	1.6	1.37	1.86	1.25	1.54	1.31	1.24
X4	Oxidation State	2	3	2	2	2	1	2	2	1	2	2	2	2
X5	Electronegativity	1	1.6	1.8	1.9	1.7	0.8	1.2	1.5	0.9	1.8	1.9	1.6	1.8
X6	Covalent Radius (Å)	1.74	1.18	1.16	1.17	1.48	2.03	1.36	1.17	1.54	1.15	1.17	1.25	1.17
X7	Melting Point (°C)	839	1857	1495	1083	321	63	649	1244	98	1453	1083	419	1535
X8	Boiling Point (°C)	1484	2672	2870	2567	765	760	1090	1962	883	2732	2567	907	2750
X9	Ionic Radius (Å)	0.99	0.52	0.75	0.73	0.97	1.38	0.72	0.46	1.02	0.69	1.19	0.74	0.65
X10	Ionization Energy (kj/mol)	590	653	757	745	866	418	736	716	489	736	804	904	762
X11	Crystal Radius (Å)	1.14	0.76	0.54	0.71	0.92	1.52	0.86	0.81	1.16	0.7	1.49	0.74	0.69
X12	Electron Affinity (kj/mol)	2.4	64.3	63.7	118.4	0	48.4	0	0	53	112	35	0	15.7
X13	Atomic Volume (cm^3^/mol)	25.97	7.23	6.7	7.1	13.1	45.36	0	7.39	23.68	6.6	18.27	9.2	7.1
X14	Density (g cm^−3^)	1.55	7.19	8.9	8.96	8.65	0.86	1.74	7.43	0.97	8.9	11.4	7.14	7.86
X15	Atomic Radius/Atomic Weight	0.048	0.024	0.021	0.021	0.013	0.058	0.065	0.025	0.08	0.02	0.007	0.02	0.022
X16	Ion Charge	1.04	2.4	1.6	1.48	1.35	0.44	1.25	1.45	0.53	1.6	1.29	1.52	1.61
X17	Ion Charge/(Atomic Radius)2	0.54	1.92	1.28	1.09	0.91	0.19	0.78	1.06	0.28	1.28	0.84	1.16	1.3
X18	Standard Reduction Potential (V)	−2.87	−0.74	0	0.34	−0.4	−2.93	−2.37	0	−2.71	−0.25	−0.13	−0.76	−0.44
X19	(Atomic Radius)^2^ (Å)	3.68	1.56	1.56	1.82	2.19	5.15	2.56	1.87	3.45	1.56	2.37	1.71	1.53
X20	Cationic Polarizing Power	2.08	7.2	3.2	2.96	2.7	0.44	2.5	2.91	0.53	3.2	2.59	3.05	3.22
X21	Covalent Index	1.92	3.2	4.05	4.87	4.27	1.45	2.3	3.08	1.5	4.05	5.55	3.35	4.01

## 4. Conclusions

The study performed the adsorption of metal ions on low-cost agro-waste from metal-infested water samples. The best-suited adsorption parameters were achieved after the experimental work (60 ppm initial concentration, 60 mint contact time and 7 pH). The results were correlated with the properties of metal ions that are responsible for metal attachment on the surface of an adsorbent. A computer model based on the experimental results was generated to develop the ion characteristics and property relationship. This model not only demonstrates the adsorption of selected metal ions used during experiments but also predicts the adsorption potential of even those metals for which the experiment was not performed. Therefore, based on the generated model, we can predict the adsorption of a broad range of metal ions on selected novel adsorbents. The current study is a fruitful addition for environmental remediation with cost-effective adsorbents and future predictions through model generation.

## Figures and Tables

**Figure 1 molecules-27-08213-f001:**
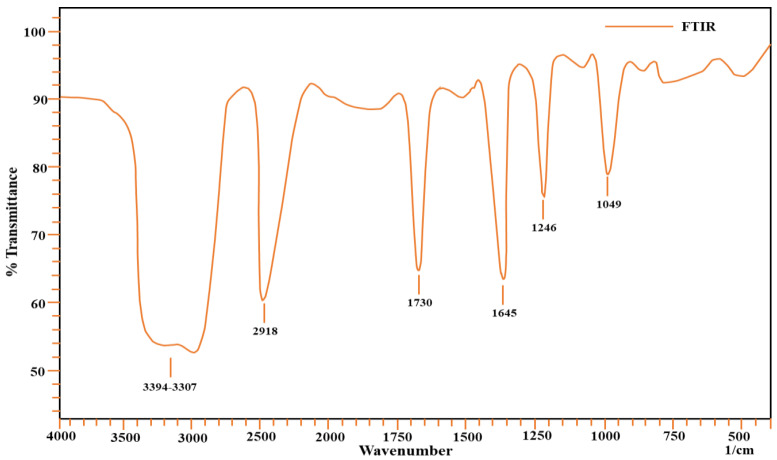
FTIR spectra of *Aerva javanica*.

**Figure 2 molecules-27-08213-f002:**
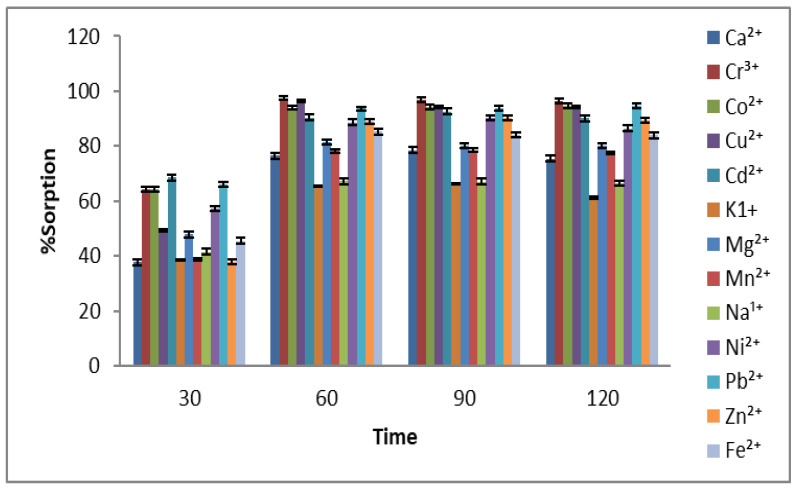
Effect of contact time (30–120 min) on % adsorption of 13 selected metal ions using *Aerva javanica* as the sorbent.

**Figure 3 molecules-27-08213-f003:**
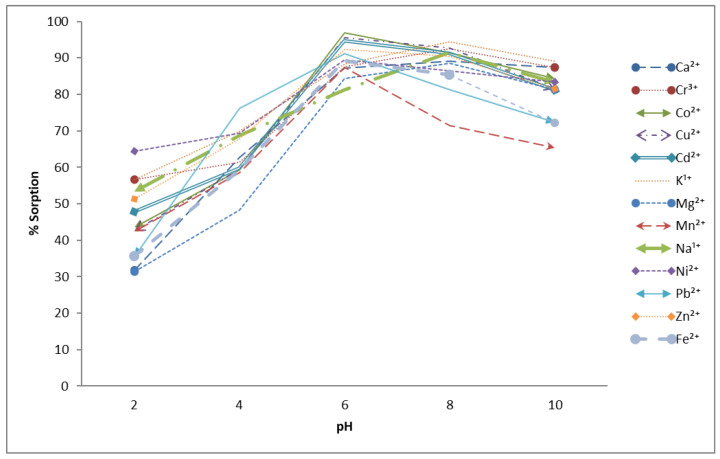
Effect of pH on % sorption of metal ions using *Aerva javanica* as the sorbent (60 ppm each metal concentration, 60 min contact time and 150 rpm shaking speed).

**Figure 4 molecules-27-08213-f004:**
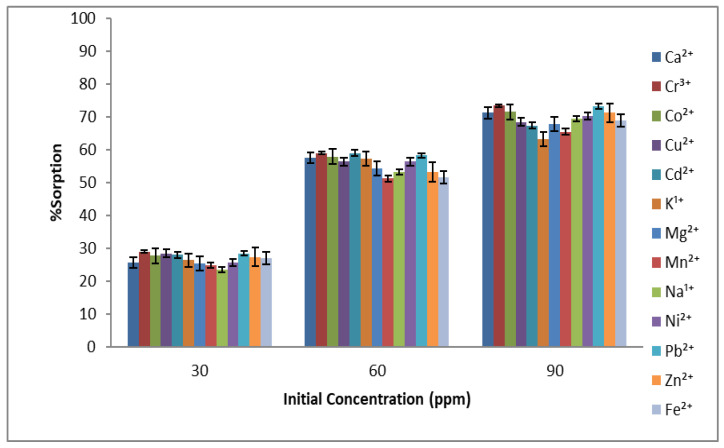
Effect of the initial sorbate concentration on % sorption of metal ions using *Aerva javanica* as the sorbent (at pH 6, 60 min time and 150 rpm shaking speed).

**Table 1 molecules-27-08213-t001:** Linear regression analysis to show the relationship between % sorption onto *Aerva javanica* and metal ionic characteristics.

Sr. No	Model Equation	R	R^2^	R^2^ Adjusted	*p*-Value	F-Value	Standard Error
1	83.48 + 0.2084 X1	0.7082	0.5016	0.4563	0.0067	11.0739	3.9694
2	84.44 + 0.0781 X2	0.6958	0.4842	0.4373	0.0082	10.3277	4.0382
3	105.94 − 10.80 X3	0.6445	0.4154	0.3623	0.0173	7.8189	4.2990
4	76.52 + 6.79 X4	0.6226	0.3877	0.3320	0.0230	6.9655	4.3999
5	71.86 + 11.81 X5	0.8592	0.7383	0.7146	0.0002	31.0478	2.8760
6	106.87 − 12.79 X6	0.6549	0.4289	0.3771	0.0151	8.2643	4.2490
7	84.17 + 0.0058 X7	0.6318	0.3991	0.3445	0.0205	7.3080	4.3585
8	81.73 + 0.0042 X8	0.6929	0.4801	0.4329	0.0086	10.1604	4.0541
9	93.6887 − 4.9411 X9	0.2424	0.0588	−0.0267	0.4247	0.6872	5.4552
10	73.46 + 0.0228 X10	0.5871	0.3448	0.2852	0.0348	5.7887	4.5515
11	93.8452 − 4.6011 X11	0.2650	0.0702	−0.0142	0.3815	0.8309	5.4219
12	88.3185 + 0.0320 X12	0.2500	0.0625	−0.0226	0.4099	0.7338	5.4443
13	94.06 − 0.278 X13	0.4717	0.2225	0.1518	0.1036	3.1484	4.9580
14	81.49 + 1.290 X14	0.8711	0.7588	0.7369	0.0001	34.6113	2.7614
15	95.90 − 191.7 X15	0.8041	0.6467	0.6146	0.0009	20.1372	3.3421
16	80.09 + 7.01 X16	0.6458	0.4170	0.3640	0.0171	7.8700	4.2931
17	82.57 + 7.19 X17	0.6182	0.3821	0.3260	0.0243	6.8050	4.4197
18	93.1696 + 3.5159 X18	0.7982	0.6372	0.6042	0.0010	19.3254	3.3865
19	97.26 − 3.21 X19	0.6525	0.4257	0.3735	0.0156	8.1566	4.2609
20	84.2105 + 1.9073 X20	0.5721	0.3274	0.2662	0.0410	5.3546	4.6115
21	76.95 + 3.762 X21	0.8956	0.8022	0.7842	3.4705	44.6147	2.5007

**Table 2 molecules-27-08213-t002:** Langmuir and Freundlich isotherm parameters for the adsorption of metal ions onto *Aerva javanica*.

Metal Ion	Langmuir Constants				Freundlich Constants		
	Q	b	R_L_	r^2^	K_F_	n	r^2^
Ca^2+^	17.9917	0.3418	0.0005	0.1178	3.2411	1.0719	0.9327
Cr^3+^	82.0546	0.1109	0.0001	0.3018	7.6718	1.0177	0.9992
Co^2+^	105.429	0.0611	0.0019	0.9929	6.4224	0.9988	0.9801
Cu^2+^	5.57382	1.9005	7.9050	0.0101	9.6783	0.9509	0.9717
Cd^2+^	38.1898	0.2302	0.0003	0.0633	7.7037	1.0106	0.973
K^1+^	12.2645	0.4853	0.0007	0.8354	2.0179	1.1296	0.9975
Mg^2+^	16.5423	0.4711	0.0005	0.7363	2.9194	1.1169	0.9969
Mn^2+^	11.2281	0.4882	0.0007	0.3414	2.4666	1.0874	0.9505
Na^1+^	15.4492	0.3542	0.0005	0.1631	3.5473	1.0426	0.9405
Ni^2+^	20.0670	0.3541	0.0004	0.1664	4.7566	1.0397	0.9639
Pb^2+^	178.043	0.0415	4.6805	0.0005	10.057	1.0004	0.8718
Zn^2+^	55.3924	0.0943	0.0002	0.0339	5.2239	0.9968	0.9803
Fe^2+^	32.7997	0.1861	0.0003	0.0488	8.8613	0.9493	0.9869

**Table 3 molecules-27-08213-t003:** Linear regression analysis to show the relationship between the Langmuir constant (Q) for *Aerva javanica* and metal ionic characteristics.

Sr. No	Model Equation	R	R^2^	R^2^ Adjusted	*p*-Value	F-Value	Standard Error
1	−3.2741 + 2.0797 X1	0.6556	0.4298	0.3780	0.0149	8.294	45.78
2	5.3064 + 0.7959 X2	0.6577	0.4325	0.3809	0.0145	8.385	45.67
3	152.3708 − 62.5182 X3	0.3460	0.1197	0.0397	0.2468	1.496	56.89
4	−16.018 + 38.3228 X4	0.3258	0.1061	0.0249	0.2772	1.306	57.32
5	−84.1921 + 94.5812 X5	0.6380	0.4070	0.3531	0.0189	7.551	46.69
6	192.2561 − 99.5727 X6	0.4729	0.2236	0.1530	0.1026	3.169	53.42
7	26.51424 + 0.03336 X7	0.3371	0.1136	0.0331	0.2599	1.411	57.08
8	−8.2801 + 0.0357 X8	0.5400	0.2916	0.2272	0.0567	4.529	51.03
9	47.01664 + 12.835 X9	0.0584	0.0034	−0.0871	0.8496	0.0376	60.53
10	−53.3461 + 0.1572 X10	0.3750	0.1406	0.0625	0.2067	1.800	56.20
11	57.6593 + 0.0221 X11	0.0002	1.390	−0.0909	0.9996	1.530	60.63
12	34.6938 + 0.5826 X12	0.4213	0.1775	0.1027	0.1516	2.374	54.99
13	70.8225 − 0.9614 X13	0.2000	0.0400	−0.0472	0.5123	0.4583	59.40
14	7.74354 + 10.4292 X14	0.6533	0.4268	0.3747	0.0154	8.192	45.90
15	104.634 − 1425.06 X15	0.5543	0.3072	0.2443	0.0493	4.879	50.46
16	4.5330 + 39.25424 X16	0.3353	0.1124	0.0317	0.2626	1.394	57.12
17	19.2438 + 39.4294 X17	0.3143	0.0988	0.0169	0.2954	1.206	57.55
18	85.6583 + 27.4340 X18	0.5776	0.3337	0.2731	0.0386	5.509	49.49
19	102.634 − 18.8146 X19	0.3544	0.1256	0.0461	0.2347	1.580	56.69
20	27.7141 + 10.6366 X20	0.2959	0.0875	0.0046	0.3262	1.055	57.92
21	−56.996 + 34.1553 X21	0.7542	0.5688	0.5296	0.0028	14.51	39.81

**Table 4 molecules-27-08213-t004:** Linear regression analysis to show the relationship between the adsorption capacity (K_F_) of *Aerva javanica* obtained from the Freundlich isotherm and metal ionic characteristics.

Sr. No	Model Equation	R	R^2^	R^2^ Adjusted	*p*-Value	F-Value	Standard Error
1	2.457681 + 0.106166 X1	0.6361	0.4046	0.3505	0.0194	7.478	2.461
2	2.938278 + 0.039985 X2	0.6279	0.3943	0.3393	0.0215	7.162	2.482
3	14.15095 − 5.66598 X3	0.5960	0.3552	0.2966	0.0315	6.062	2.561
4	0.519302 + 2.625931 X4	0.4243	0.1800	0.1055	0.1483	2.416	2.888
5	−4.03748 + 6.404432 X5	0.8211	0.6742	0.6446	0.0005	22.77	1.821
6	14.76476 − 6.8038 X6	0.6142	0.3772	0.3206	0.0255	6.664	2.517
7	3.503003 + 0.002213 X7	0.4248	0.1805	0.1060	0.1478	2.423	2.888
8	1.942227 + 0.001964 X8	0.5644	0.3185	0.2566	0.0444	5.143	2.633
9	6.86076 − 1.55469 X9	0.1344	0.0180	−0.0711	0.6613	0.2026	3.161
10	−3.14565 + 0.012347 X10	0.5595	0.3130	0.2506	0.0467	5.013	2.644
11	7.645199 − 2.24156 X11	0.2275	0.0518	−0.0344	0.4545	0.6009	3.106
12	4.628833 + 0.023834 X12	0.3276	0.1073	0.0261	0.2745	1.323	3.014
13	6.926693 − 0.09931 X13	0.3926	0.1541	0.0773	0.1844	2.006	2.934
14	1.390868 + 0.666069 X14	0.7930	0.6289	0.5952	0.0012	18.65	1.943
15	8.643302 − 93.2991 X15	0.6898	0.4758	0.4282	0.0090	9.986	2.309
16	1.203276 + 3.224654 X16	0.5236	0.2742	0.2082	0.0662	4.156	2.718
17	2.109756 + 3.548835 X17	0.5378	0.2892	0.2246	0.0579	4.477	2.689
18	7.417478 + 1.812341 X18	0.7253	0.5261	0.4830	0.0050	12.21	2.196
19	9.541114 − 1.66236 X19	0.5951	0.3542	0.2955	0.0318	6.034	2.563
20	3.163031 + 0.85409 X20	0.4516	0.2040	0.1316	0.1212	2.819	2.846
21	−1.33784 + 2.057201 X21	0.8634	0.7455	0.7224	0.0002	32.23	1.609

## Data Availability

The data are available in the manuscript.

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
