# Peer review of "Development of Ion Character Property Relationship (IC-PR) for Removal of 13-Metal Ions by Employing a Novel Green Adsorbent Aerva javanica"

_molecules, 2022, doi:10.3390/molecules27238213_

Round 1

Reviewer 1 Report

Manuscript ID: molecules-1961445

Manuscript title: Development of Ion Character Property Relationship (IC-PR) For Removal of 13-Metal ions by employing a novel Green adsorbent Aerva javanica

The authors proposed a mathematical model for study the removing of metal ions from aqueous solutions by using “green adsorbent Aerva javanica”. The calculation is based on a few of atomic and ionic properties of metals. The idea of correlating the properties of ions with the ability to be adsorbed is not new, many other studies have been published, including mathematical models. In the proposed model, the properties of atoms were mixed with those of ions and bulk metals, although the adsorption is about only ions. In addition, the manuscript is not well written, having drafting and interpretation mistakes. Therefore, I cannot propose the manuscript publication in "Molecules".

I have the following observations:

1.     It is not clear if the experimental results were done by the authors or taken from the literature. The experimental data for the adsorption study are missing.

2.     The mathematical calculation must have a chemical basis, must be correlated with chemical properties and the correlations must be well explained.

3.     A few sentences from the abstract must be reformulated like “can be lead absorbent for the removal of heavy metals ions”, “The merits and advantages of using this adsorbent are frequently available in Pakistan, cost effective, environmentally benign processing and acted as versatile broad range adsorbent for different metals ions.”

4.     In Abstract, the information about the experiment (“The optimization re-20 mediation parameters of metal ions for batch adsorption approach were developed which were 21 initial concentrations (60 ppm), contact time (60 minutes) and pH (7).”) can not be found in manuscript. Where is it from?

5.     In Abstract – “therefore 21 properties of metal ions were selected which are called as “descriptors”” - the chosen properties are not only those of metal ions, and some of them are unrepresentative (e.g. boiling point of metals).

6.     In “Introduction” – the explanation about “natural resources” is redundant at this level. Other well-known platitudes can be found in the introduction. The introduction must be seriously revised, with concrete information strictly related to the subject of the manuscript.

7.     In “Results & Discussion” some information is repeating. Also, the text should be more concise.

8.     The FTIR spectrum and interpretation must be correlated with the other experimental data, not to remain isolated. Concretely, what is the information from FTIR useful for?

9.     The band assigned to -OH is not broad. The transmittance values on the axis are missing.

10.  “The –OH group and heteroatoms play an important role in attaching the sorbate to the surface for adsorption” – An explanation is missing.

11.  The definition of covalent index for an atom is missing.

12.  Too much theory can be found in the section “Results & Discussion”.

13.  Many unnecessary details can be found in the section “3. Materials & Methods”.

14.  Some information is repeated. For example in 3.3 – “Fourier Transform Infrared Spectroscopy (FTIR) was performed for functional group analysis on the surface of adsorbent” and 2.1 - “Using Fourier transform infrared spectroscopy functional group analysis of the biosorbent were analyzed”; in 2.1.1 - “The surface of the adsorbent was examined using a Fourier Transform Infrared Spectrometer (Model Shimadzu AIM-8800)” and 3.4 - “Fourier Transform Infrared Spectrophotometer was used to identify functional groups in the sorbent's structure (Model Shimadzu AIM-8800).”

15.  The method used for the metal determination must be described more precisely if the experiments were done by the authors.

16.  The measure units are not given in Table 5.

17.  A few of properties from Table 5 are for atoms or bulk metals (e.g. density, melting and boiling point, etc.), but the adsorption study is about the ions’ adsorption. For example, which can the correlation between the density, the melting or boiling point of chromium (bulk metal) and the adsorption of Cr(III) ions?

18.  “Results were correlated with properties of metal ions which are responsible for metal attachment on the surface of adsorbent” – Which are these properties and how be explained the correlation?

19.  The references for the information in Table 5 are missing.

Author Response

RESPONSE LETTER-1

Manuscript molecules-1961445

Development of Ion Character Property Relationship (IC-PR) For Removal of 13-Metal ions by employing a novel Green adsorbent Aervajavanica

 Dear Editor and Reviewer-1

Authors are again thankful to editorial office and reviewer-1 for suggestions in form of revisions to improve the paper. All suggestions are properly incorporated in paper. Reviewer Comments are given in italic while response to reviewer comments is given in plane text. All additions made in main manuscript are in track changes.

Thanks and regards

Authors

Reviewer 2 Report

1- The text is not sufficiently clear and the use of English should be improved,

2-Presentation of the results must be improved.

3- It is difficult to figure out the important features of figure 1( numbers, axes captions, text...etc). Please, provide high-resolution Figures. 

4- please provide a section for methods and modelling

5- Re-write the Introduction to be organized, succinct, clear, and informative. 

6- Add  a paragraph in the introduction to show the author's motivation and novelty of this work 

7- Revise all equations and follow up the style and guidelines of the journal

8- Provide the obtained key results in the conclusion

9- Please correlate the obtained results and provide more discussions

Author Response

RESPONSE LETTER-2

Manuscript molecules-1961445

Development of Ion Character Property Relationship (IC-PR) For Removal of 13-Metal ions by employing a novel Green adsorbent Aervajavanica

 Dear Worthy editor and Reviewer-2

Authors are thankful to editorial office and reviewer-2 for suggestions in form of revisions to improve the paper. All suggestions are properly incorporated in paper. Reviewer Comments are given in italic while response to reviewer comments is given in plane text. All additions made in main manuscript are in track changes.

Thanks and regards

Authors

Reviewer 3 Report

In the present manuscript, the novel Aera javanica absorbent was applied for the removal of metal ions. The manuscript can be accepted once the following queries are being satisfactorily addressed and/or clarified in the revision. The detailed questions are listed below:

1. More physical and chemical properties and photos of desert cotton should be shown.

2. Excessive use of flattening in data processing for Figure 1. Is the raw data shown in the figure, and is it smoothed?

3. The structure of the article is not clear, and the experiment should be placed before the results and discussion.

4. It is recommended to use more intuitive pictures to present, instead of just listing numbers in tables.

Author Response

RESPONSE LETTER-3
Manuscript molecules-1961445
Development of Ion Character Property Relationship (IC-PR) For Removal of 13-Metal
ions by employing a novel Green adsorbent Aerva javanica

Dear Editor and Reviewer-3

Authors are again thankful to editorial office and reviewer-3 for suggestions in form of revisions to improve the paper. All suggestions are properly incorporated in paper. Reviewer Comments are given in italic while response to reviewer comments is given in plane text. All additions made in
main manuscript are in track changes. 

Thanks and Regards

Ali Irfan

Round 2

Reviewer 2 Report

Can be accepted in the current version

Author Response

REVIEWER-2

Dear worthy Reviewer-2

Greetings! Thanks for your comments to improve the quality of the manuscript.
Dear, we modified the attached manuscript according to your comments. I hope this manuscript will fulfill the criteria of acceptance.

Thanks and regards

Ali Irfan
